# Enlightenment on oscillatory properties of 23 class B notifiable infectious diseases in the mainland of China from 2004 to 2020

Chuanliang Han[1☯]*, Meijia Li[2☯], Naem Haihambo[2], Yu Cao[3], Xixi Zhao[4,5,6]*

**1** State Key Laboratory of Cognitive Neuroscience and Learning & IDG/McGovern Institute for Brain Research, Beijing Normal University, Beijing, China, **2** Faculty of Psychology and Center for Neuroscience, Vrije Universiteit Brussel, Brussels, Belgium, **3** State Key Laboratory of Earth Surface Process and Resource Ecology and Ministry of Education Key Laboratory for Biodiversity Science and Ecological Engineering, College of Life Sciences, Beijing Normal University, Beijing, China, **4** Beijing Anding Hospital, Capital Medical University, Beijing, China, **5** The National Clinical Research Center for Mental Disorders & Beijing Key Laboratory of Mental Disorders, Beijing Anding Hospital, Capital Medical University, Beijing, China, **6** Advanced Innovation Center for Human Brain Protection, Capital Medical University, Beijing, China

☯ These authors contributed equally to this work.
\* zhaoxixi@ccmu.edu.cn (XZ); hanchuanliang2014@163.com (CH)

**Data Availability Statement:** The data was obtained from the National Health Commission of China (http://www.nhc.gov.cn/), which is totally

## Abstract

A variety of infectious diseases occur in mainland China every year. Cyclic oscillation is a widespread attribute of most viral human infections. Understanding the outbreak cycle of infectious diseases can be conducive for public health management and disease surveillance. In this study, we collected time-series data for 23 class B notifiable infectious diseases from 2004 to 2020 using public datasets from the National Health Commission of China. Oscillatory properties were explored using power spectrum analysis. We found that the 23 class B diseases from the dataset have obvious oscillatory patterns (seasonal or sporadic), which could be divided into three categories according to their oscillatory power in different frequencies each year. These diseases were found to have different preferred outbreak months and infection selectivity. Diseases that break out in autumn and winter are more selective. Furthermore, we calculated the oscillation power and the average number of infected cases of all 23 diseases in the first eight years (2004 to 2012) and the next eight years (2012 to 2020) since the update of the surveillance system. A strong positive correlation was found between the change of oscillation power and the change in the number of infected cases, which was consistent with the simulation results using a conceptual hybrid model. The establishment of reliable and effective analytical methods contributes to a better understanding of infectious diseases' oscillation cycle characteristics. Our research has certain guiding significance for the effective prevention and control of class B infectious diseases.

## Introduction

Infectious diseases are a type of disease caused by various pathogens, which can spread among people and animals [1–4]. Pathogens causing infectious diseases include viruses, rickettsia,

open to the public The minimal dataset underlying the results described in our manuscript can be found https://github.com/Stellapros/Dataset-of-infectious-disease.

**Funding:** This study was sponsored by the research on prevention and control of major chronic non-communicable diseases in the Ministry of Science and Technology (2016YFC1306100), Beijing Municipal Hospital Clinical Technology Innovation and Research Plan (XMLX201805), Beijing Municipal Hospital Research and Development Project (PX2021068).

**Competing interests:** The authors have declared that no competing interests exist.

mycoplasma, bacteria, fungi, parasites, etc [5,6]. The pathological process of infectious diseases depends on the nature of pathogenic microorganisms and the body's response thereto, as well as timely and appropriate treatment [7–9]. Most infectious diseases can be cured by strengthening the body's resistance to the appropriate pathogen and proper treatment [10,11]. If the body's immune resistance is poor and the infection is not treated on time, the infection may become chronic or spread, or may result in death [12,13].

Apart from the impact of infectious diseases on individuals, the outbreak of infectious diseases can be periodic. Recurrence is a common feature of infectious diseases [14] that has been confirmed in many countries around the world [15,16]. Examples of these are the seasonal pertussis patterns such as measles [17] in Europe [18–22], influenza [23] in Japan, and measles [23] and rabies [24–26] in China. This oscillation may be driven by natural factors, such as seasonal temperature, rainfall [27,28], natural disasters [29], or human factors, such as school terms [30,31], economic migration [32,33], or vaccination coverage [34]. It is essential to forecast the recurrent outbreaks of these infectious diseases due to their global reach, impact on individual livelihoods, as well as on the economy [35,36] and public mental health systems [37,38]. For example, in mainland China, in the period between January 1st to December 31st, 2019, 10244507 cases of notifiable infectious diseases were reported in total, with 25285 resulting in death. Understanding cyclical outbreaks of seasonal or sporadic epidemics plays an important role in epidemic prevention and control [14,39].

To better assess and control epidemic outbreaks, the Chinese government strengthened the country's infectious disease surveillance system [40] after the outbreak of severe acute respiratory syndrome (SARS) in 2003. The infectious diseases in this system were divided into notifiable classes A, B, and C. Class A notifiable diseases like the plague and cholera can cause large-scale, severe epidemics within a short period of time. Class B notifiable diseases like AIDS and Anthrax may cause moderate epidemic outbreaks. Class C notifiable diseases like rubella and conjunctivitis are less severe and less infectious, causing mild outbreaks. The rate of infection of class A infectious diseases in China is very low, which suggests that it has been well controlled in China. Class B infectious diseases are not only highly infectious, but also have higher mortality than those of class C infectious diseases. Therefore, the study of infectious characteristics of class B infectious diseases is very important. Previous studies on infectious diseases that happened in China, however, are largely ignored. Of these few studies, investigations are mainly concentrated on one or a small number of infectious diseases, or only focus on a short time period. We are still far from having a concise method of analysis that can account for both the annual incidence patterns of infectious diseases in humans and the evolution of the diseases.

In this study, we first illustrated the time series of infected cases from 23 class B infectious diseases in mainland China from 2004 to 2020 and conducted a power spectrum analysis on this data. Based on different spectrums, we were able to categorize the diseases and subsequently investigate the preferred month of each infectious disease in a year. Further, we analyzed the correlation between the change in oscillation power of the infectious diseases and the rate of change of infected cases in the first eight years (2004 to 2012) and the last eight years (2012 to 2020) since the update of the surveillance system. Finally, we summarized the main findings as a table and established a conceptual model to illustrate the mechanism of the oscillatory characteristics.

## Methods

### Data and sources

Available time series data for the monthly reported and confirmed cases of 23 class B notifiable infectious diseases in China's mainland, from April 2004 to September 2020, was obtained

**Table 1. Summary of the main finding in 23 infectious diseases in mainland China.**

| Name of Infectious Diseases | Type | Preferred Month | Selectivity | Changes in the number of infected cases from first 8 years to second 8 years (log) | Changes in the power of infected cases from first 8 years to second 8 years (log) |
|---|---|---|---|---|---|
| HIV | 3 | Dec | 0.56 | -1.11 | -0.92 |
| HAV | 1 | Aug | 0.38 | 1.02 | 2.48 |
| HBV | 3 | Mar | 0.23 | 0.08 | 0.09 |
| HCV | 3 | Mar | 0.3 | -0.66 | -0.78 |
| HEV | 1 | Mar | 0.5 | -0.25 | 0.4 |
| Measles | 1 | May | 0.87 | 1.35 | 2.11 |
| Haemorrhagic fevers, Viral | 2 | Nov | 0.78 | 0.22 | 0.56 |
| Rabies | 1 | Oct | 0.51 | 1.27 | 2.32 |
| Japanese Encephalitis | 1 | Aug | 0.998 | 1.16 | 2.61 |
| Dengue & Severe Dengue | 1 | Oct | 0.995 | -3.48 | -7.71 |
| Anthrax | 1 | Aug | 0.89 | 0.31 | 0.79 |
| Shigella Species or Entamoeba histolytia | 1 | Aug | 0.81 | 0.99 | 2.59 |
| Tuberculosis | 1 | Mar | 0.28 | 0.24 | 0.75 |
| Typhoid Fever & Paratyphoid Fever | 1 | Aug | 0.68 | 0.68 | 2.05 |
| Pertussis | 1 | Aug | 0.66 | -1.29 | -2.92 |
| Neonatal Tetanus | 1 | Aug | 0.34 | 1.99 | 3.2 |
| Scarlet Fever | 2 | Dec | 0.78 | -0.59 | -1.2 |
| Brucellosis | 1 | Jun | 0.68 | -0.5 | -0.31 |
| Gonorrhea | 1 | Aug | 0.37 | 0.24 | 0.66 |
| Treponema Pallidum | 1 | Jul | 0.33 | -0.6 | -0.82 |
| Leptospirosis | 1 | Sep | 0.98 | 0.89 | 2.5 |
| Schistosomiasis | 1 | Oct | 0.87 | -0.82 | -4.64 |
| Malaria | 1 | Aug | 0.88 | 2.29 | 7.04 |

from the National Health Commission of China (http://www.nhc.gov.cn/). The dataset is open to the public, reported by the Chinese Centre for Disease Control and Prevention (CDC) each month. These diseases are AIDS (HIV), hepatitis disease (including Hepatitis A virus (HAV), Hepatitis B virus (HBV), Hepatitis C virus (HCV), and Hepatitis E virus (HEV)), Measles, Haemorrhagic fevers, Dengue, and severe dengue, Rabies, Japanese encephalitis, Anthrax, Shigella species or Entamoeba histolytica, Tuberculosis, Typhoid fever & Paratyphoid fever, Pertussis, Neonatal Tetanus, Scarlet fever, Brucellosis, Gonorrhea, Treponema pallidum, Leptospirosis, Schistosomiasis, and Malaria (Table 1, 1st column). The data sampling rate is one point per month (12 time points per year) by the monthly report of the National Health Commission of China.

## Spectrum analysis

To better quantify the oscillatory property of each infectious disease, we used the spectrum analysis. Similar methods have been used in classic and modern studies in the field of infectious diseases [6,15,17,19,41]. Spectrum analysis is a technique for decomposing complex signals into simpler signals based on the Fourier transform. Many biological signals can be expressed as the sum of various simple signals of different frequencies and produce information of a signal at different frequencies (such as amplitude, power, intensity, or phase, etc.).

The power spectral density (PSD) for each infectious disease during these 16 years was computed using the multi-taper method with five tapers using the Chronux toolbox [42] an

open-source, data analysis toolbox (Chronux) available at http://chronux.org. Power spectra of the time series data of infected cases of each disease was calculated from 2004 to 2020. Essentially, the multi-taper method attempts to reduce the variance of spectral estimates by pre-multiplying the data with several orthogonal tapers known as Slepian functions. The frequency decomposition of multi-tapered data segments provides a set of independent spectral estimates that, once averaged, yield a more reliable ensemble estimate of noisy data.

## Classification of different clusters of diseases

We noticed that we could distinguish different infectious diseases by the number of outbreaks in a year. To classify the different clusters of infectious diseases based on their oscillatory characteristics, we used two features: the power ratio between once a year and twice a year, and the power ratio between once a year and three times a year. The definition of power ratio is the ratio between the powers corresponding to two different frequencies (times per year). We then set two linear thresholds that precisely separated them into three clusters.

## Tuning curves for monthly infected cases

We assumed that all infectious diseases included in this study have a similar trend each year in the 16 years of observation. Based on this assumption, we took the monthly average number of infected cases- during all 16 years and computed them into a tuning curve. Each infectious disease in this study has a tuning curve, and the oscillatory pattern within a year is clear.

## Preferred month and selectivity of the epidemic outbreak

After getting the tuning curve of each disease, we aimed to better capture the property of oscillations for infectious diseases in a year. Two indices were defined: preferred month and infection selectivity.

The preferred month index is defined as the month in a year that has the most cases of infections. The infection selectivity index is defined as 1 minus the ratio of the minimum and the maximum number of infected cases in a year. If the selectivity index is closer to 1, then the shape of the tuning curve is sharper, and vice versa.

## Correlation analysis

We used the Spearman correlation to measure the relationship between the selectivity index and the preferred month index. The Pearson correlation was used in the correlation analysis between the change in infected cases and change in oscillation power of the infectious diseases on all 23 infectious diseases.

## Conceptual hybrid model

We constructed a conceptual model to illustrate the underlying mechanism, i.e., the relationship between the change in infected cases and the change in oscillation power of infectious diseases.

The time series can be dissected into two components: trend component (TC) that can be modeled as a monotonically increasing function, and oscillatory component (OC) that can be modeled as a sine function. The multiplication of these two components constitutes the multiplication mechanism. The addition of these two components constitutes the additive mechanism. The hybrid mechanism combined addition and multiplication.

We then simulated the time series using this conceptual model by adding Gaussian noise (mean = 0, std = 1) to test the relationship between the change in oscillation power and the

change in the number of infected cases. The TC was simplified as a linear function and OC was simplified as a trigonometric sine function.

## Model fitting and evaluation

We further fitted the time series data of 23 infectious diseases using these three models respectively, which are shown in Eqs 1–3. The additive model is the summation of a trend component and an oscillatory component (Eq 1), the multiplication model is the multiplication of a trend component and an oscillatory component (Eq 2) and the hybrid model combines the two previous models (Eq 3).

$$I(t) = \frac{A}{1 + e^{k(t-t_0)}} + B \cdot sin(2\pi ft + \varphi) + C \tag{1}$$

$$I(t) = \frac{A}{1 + e^{k(t-t_0)}} \cdot B \cdot sin(2\pi ft + \varphi) + C \tag{2}$$

$$I(t) = \frac{A}{1 + e^{k(t-t_0)}} + \frac{B \cdot sin(2\pi ft + \varphi)}{1 + e^{k(t-t_0)}} \cdot + C \tag{3}$$

Where A, k and $t_0$ represents the maximum infected cases, increasing rate and semi-saturation period of the trend component respectively, and B, f, $\varphi$ represents the amplitude, frequency and initial phase of the oscillatory component, C is the baseline of the model. The goodness of fit for the above models is defined in Eq 4. All three models have the same number of parameters, so it is fair to compare the goodness of fit amongst them.

$$\text{Goodness of fit} = 1 - \frac{\sum_{t=1}^{n} \left( R_{data}(t) - R_{model}(t) \right)^2}{\sum_{t=1}^{n} \left( R_{data}(t) - \frac{\sum_{t=1}^{n} R_{data}(t)}{n} \right)^2} \tag{4}$$

Where $R_{data}(t)$ and $R_{model}(t)$ represent the real and fitted data of the number of infected cases for a specific disease in time point t respectively, while n is the total number of the data points.

## Results

Over the past 16 years, there are clear oscillatory patterns in infectious diseases' time series in mainland China (Fig 1A). The 16-year dataset makes the tuning curve of infected cases in different months visible (Fig 1B). We were able to estimate the power spectrum in the frequency band between 0 to 6 times per year (since the sampling rate of the data is 12 data points per year, with one data pint representing one month; Fig 1C).

## Three clusters of the oscillatory patterns of the infectious diseases

It is clear that all 23 infectious diseases have had obvious patterns of oscillation in these 16 years (from 2004 to 2020). To better interpret the periodic properties throughout a year (i.e., whether the peak of the outbreak has seasonal preferences), we took the average of all 16 years' data (number of infected cases are represented as grey dots in Figs 1B and S2) to each month as a tuning curve (represented as black curves in Figs 1B and S2). Through the power spectrum analysis (Fig 1C), we found that all 23 infectious diseases have at least one clear oscillatory peak in their spectrum (S3 Fig).

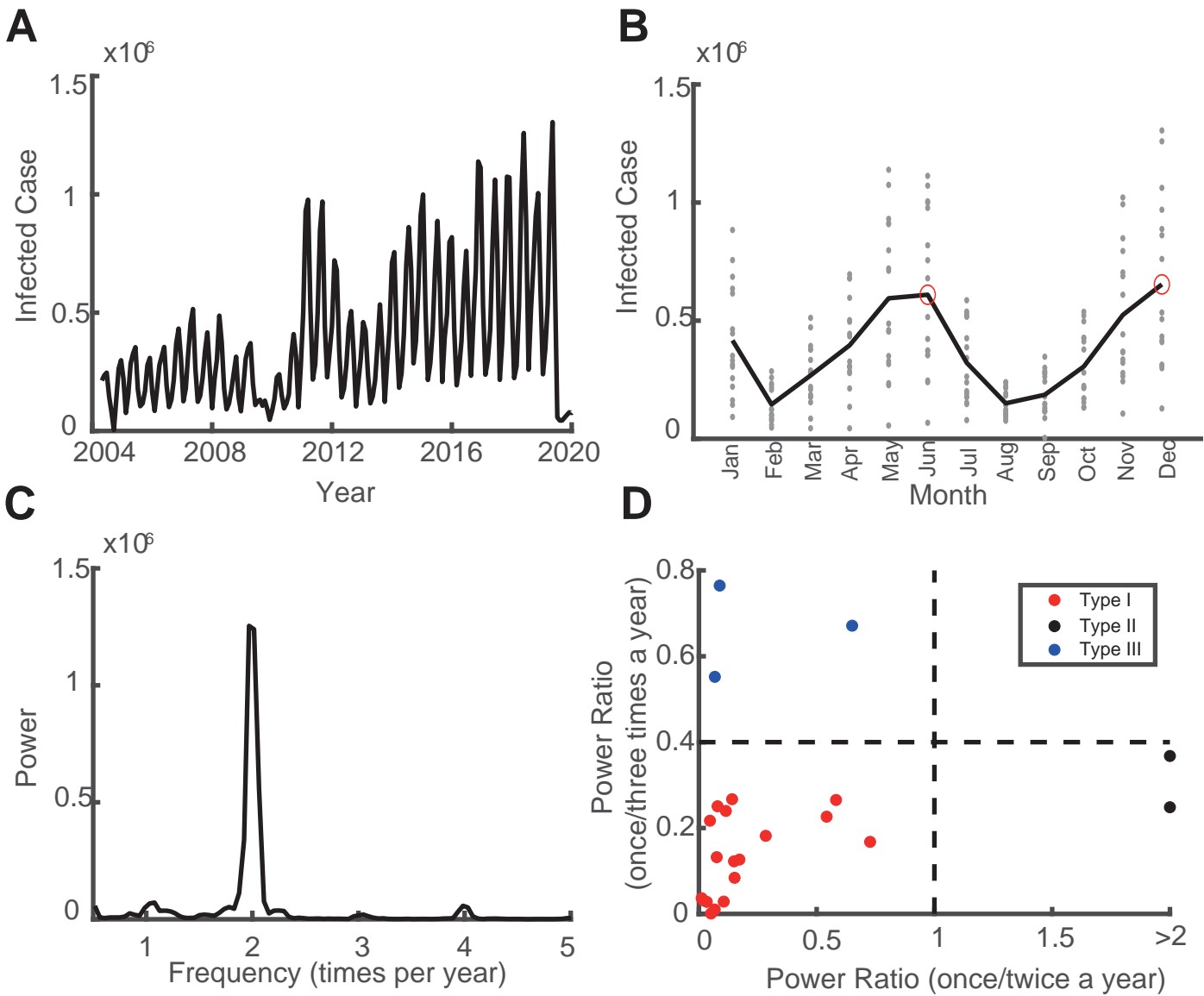

**Fig 1. Representative infectious disease with clear oscillation pattern and three oscillatory types.** Plot A shows an example time series of monthly infected cases from 2004 to 2020. In plot B, the grey dots indicate the number of infected cases every month in each year. The black curve is the average value for all years. The red circles are the peaks of the tuning curve. Plot C illustrates the power spectrum calculated from the data of left column. Plot D illustrates three clusters (denoted by red, black, and blue dots). The X-axis denotes the power ratio of occurrence between once a year and twice a year. The Y-axis denotes the power ratio of occurrence between once a year and three times a year. The dashed line is the criteria that separates them. The dots in the lower left depict diseases classified as Type I. The dots in the lower right corner depict diseases classified as Type II. The dots in the upper left corner are classified as Type III.

We then quantified the oscillatory characteristics of different diseases, and found three distinct clusters, which are illustrated in Fig 1D (similar to observations in Fig 1). The horizontal axis of this panel denotes the power ratio between once a year and twice a year, and the vertical axis denotes the power ratio between once a year and three times a year. The larger the value of the horizontal axis is, the stronger the oscillation is twice a year. The larger the value of the vertical axis is, the stronger the oscillation is three times a year. Then we set two thresholds that precisely separated them into three clusters (dashed line in Fig 1D). In total, 18 out of 23 diseases belong to Type I, two out of 23 diseases belong to Type II, the remaining three diseases belong to Type III (S4 Fig).

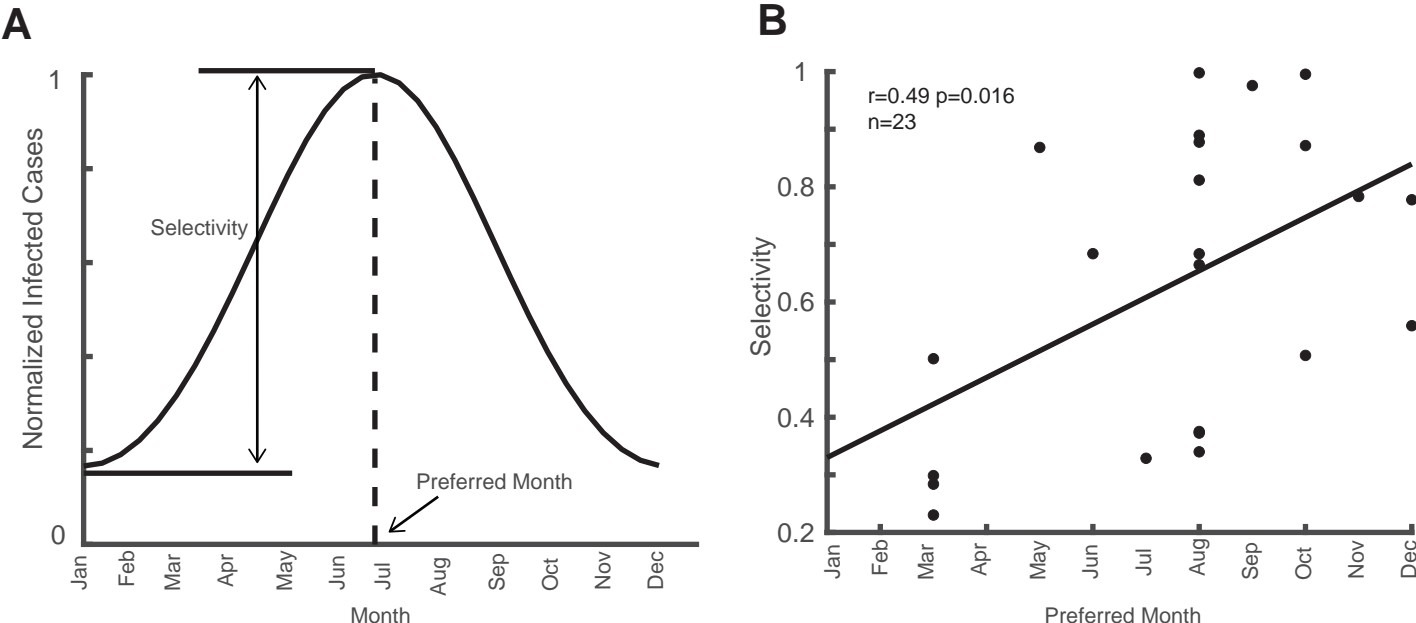

**Fig 2. Relationship between the preferred month and the selectivity index.** Plot A demonstrates preferred month and selectivity. The scatter plot in B shows the relationship between the preferred month and the selectivity index of 23 infectious diseases.

### Infectious diseases that break out in autumn and winter are more selective

Two indices (Preferred month and selectivity) were defined to capture the property of oscillations for each infectious disease in a year (Fig 2A). The preferred month is the month in a year with the most infected cases and the selectivity is the infection selectivity defined as 1 minus the ratio of minimum number and maximum number of infected cases in a year. The basic information related to the oscillatory properties helps us better understand the time and extent of their outbreak.

Furthermore, we found a significant positive correlation between the selectivity index and the preferred month index (r = 0.49, p = 0.016, Spearman correlation) (Fig 2B). In China, spring season occurs between the months of March and May, summer is from June to August, autumn is from September to November, and winter is from December to February. Hence, this significant correlation means that the outbreak of the infectious diseases in autumn and winter have a higher selectivity, while outbreaks in spring or summer tend to have more infected cases throughout the year. This provides general guidance for the prevention of different types of infectious diseases.

### Positive correlation between the change of infected cases and change of oscillatory power

We have shown the different seasonal oscillatory properties of 23 infectious diseases with static analysis. Next, we split the 16-year dataset into two parts: the first eight years (2004–2012) and the last eight years (2012–2020). In these 16 years, the number of infected cases of 14 out of 23 infectious diseases decreased over time (Fig 4A Left panel for a typical example), and nine out of 23 increased (Fig 4B Left panel for a typical example, Table 1, 5th column). This information is summarized in the 5th column of Table 1.

We then explored the relationship between the change in the number of infected cases and the corresponding strength of oscillatory power. To this end, we calculated the power

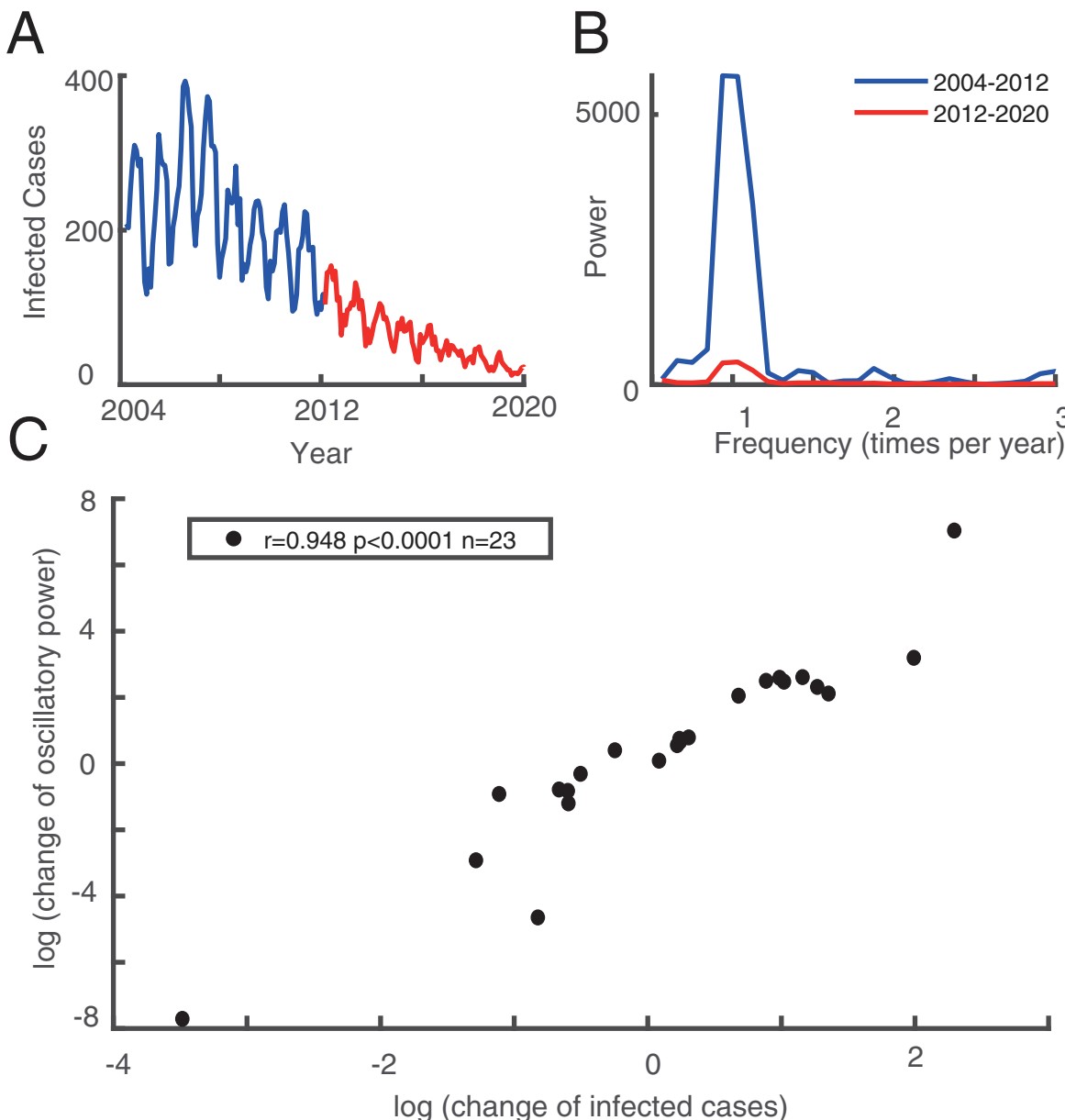

**Fig 3. Relationship between the infection and its oscillatory strength.** Plot A is an example of a disease's time series monthly infected cases from 2004 to 2020. The blue curve shows the time series of the first eight years (2004–2012) and the red curve shows the time series of the last eight years (2012–2020). Plot B shows the power spectrum calculated in first eight years (blue curve) and the last eight years (red curve). Plot C shows the scatter plot of change in mean infected cases and change in oscillatory power.

spectrums in two time periods (2004–2012 and 2012–2020) for all 23 infectious diseases. The change in the number of infected cases is defined as the ratio of the mean infected cases each month between 2004–2012 (Fig 3A and 3B, blue curve) and 2012–2020 (Fig 3A and 3B, red curve), and the change in the oscillatory power is defined as the ratio of the average power spectrum between 2004–2012 and 2012–2020. We then performed a correlation analysis between the change in infected cases and change in oscillation power of the infectious diseases on all 23 infectious diseases. We found that there is a strong positive correlation (Fig 3C)

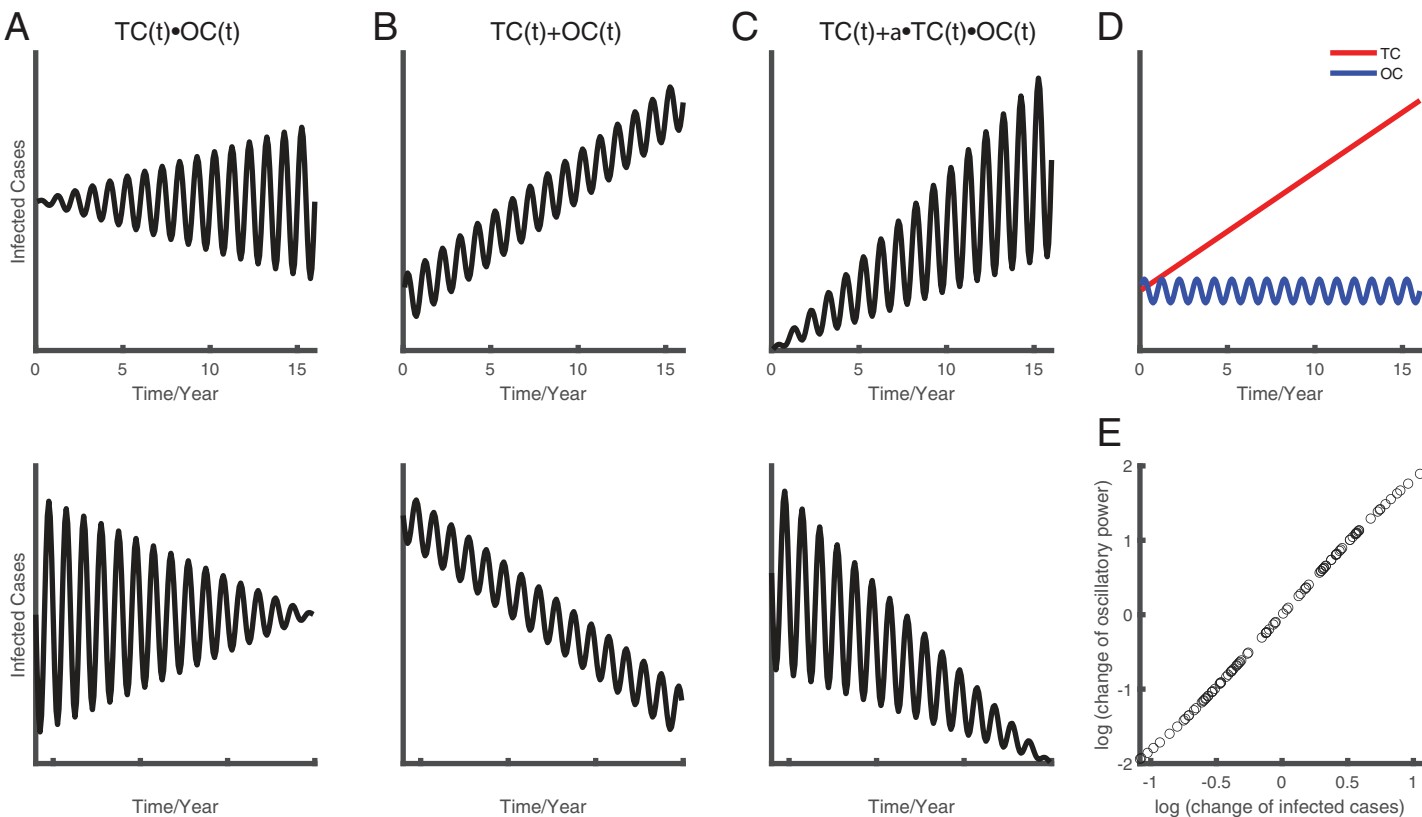

**Fig 4. The conceptual hybrid model to illustrate the oscillatory mechanism.** Three hypotheses are shown in A-C. Plot A illustrates the multiplication mechanism of tendency component (TC) and oscillatory component (OC). Plot B illustrates the additive mechanism. Plot C illustrates the hybrid mechanism. Plot D is an example of of TC and OC. E shows the result of a simulation based on the hybrid mechanism, which is consistent with results of real data in Fig 3C.

($r = 0.95$, $p < 0.0001$, Pearson correlation). This illustrates that the increase in oscillation strength often accompanies the increase in the number of infected cases.

## Hybrid model well explained the observed data

By comparing the first eight years (2004–2012) and the last eight years (2012–2020) of the available surveillance data, we can clearly see a trend in epidemic changes. The aggravation of an epidemic is not only illustrated in the increase of absolute value but also accompanied by stronger oscillation intensity. It is worth noting that this result is not inevitable since there are also other possible outcomes for time series data (Fig 4). It could also be possible that there is no correlation between the change in infected cases and the change in oscillation power of infectious diseases, which are shown as two forms: multiplication and addition mechanism. The time series can be dissected into two components: trend component (TC) that can be modeled as a linear function (Fig 4D red curve) and oscillatory component (OC) that can be modeled as sine function (Fig 4D blue curve). The multiplication of these two components constitutes the multiplication mechanism (Fig 4A). The mean infected cases remained unchanged, while the oscillatory strength increases (Fig 4A top) or decreases (Fig 4A bottom) as time goes on. The addition of these two components constitutes the multiplication mechanism (Fig 4B). As time goes on, the oscillatory strength remained unchanged, while the mean number of infected cases increases (Fig 4B top) or decreases (Fig 4B bottom).

The hybrid mechanism combined the addition and multiplication of trend and oscillatory components (Fig 4C). As time goes by, the trend of oscillatory strength and the mean number

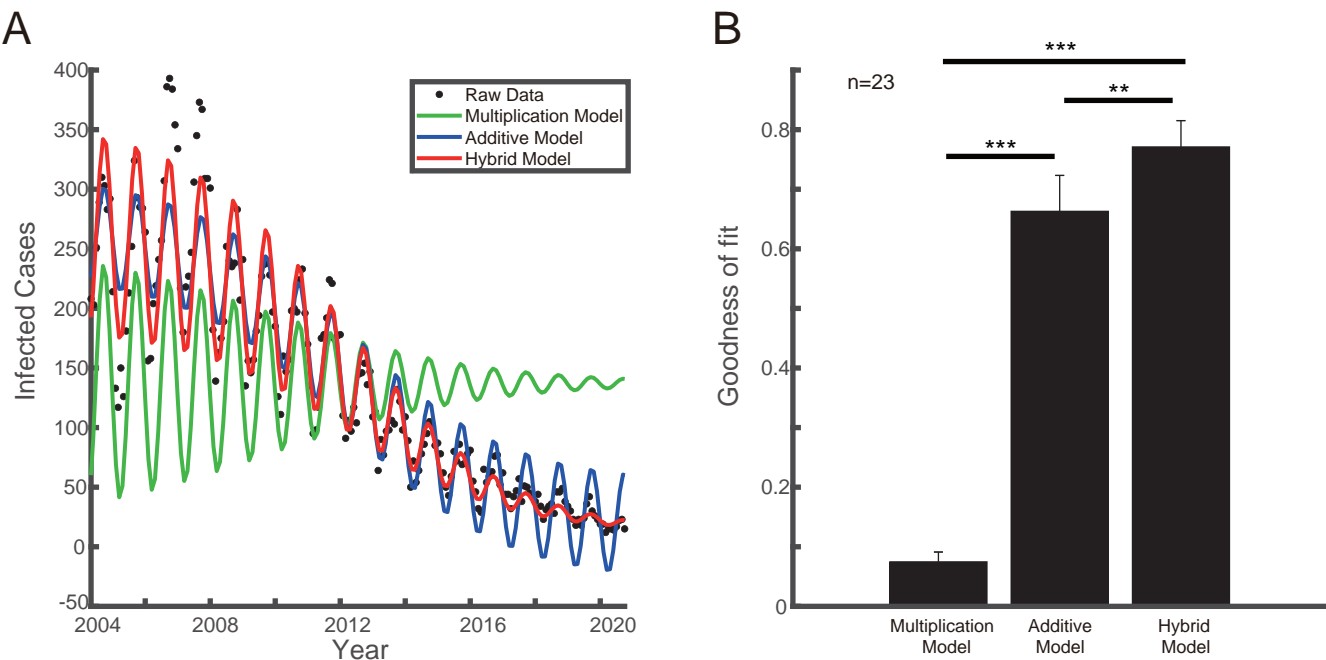

**Fig 5. Model fitting and evaluation of three models.** Plot A shows a fitting demonstration. Black dots represent the real observed data, and the solid lines in different colors are the models´ fitting curves (green for multiplication model, blue for additive model and red for hybrid model). Graph B shows the comparison of the fitting goodness of the three models (** represents for $p < 0.01$, *** represents for $p < 0.001$).

of infected cases increases (Fig 4C top) or decreases (Fig 4C bottom) together. The TC was then simplified as a linear function and OC was simplified as a trigonometric sine function (Fig 4D). We then simulated this conceptual model by adding some noise to test the relationship between the change in oscillation power and the change in the number of infected cases, which is positively correlated. This relationship was consistent with the results of the analysis using real data (Fig 3C).

To further test the hybrid hypothesis, we fit the observed data using the three models (addition, multiplication, and hybrid) for each disease (Fig 5A). We found that the goodness of fit for the hybrid model is significantly larger than that of the other two models against the hybrid hypothesis (t test with Bonferroni correction, Fig 5B). Hence, based on the real analysis from mainland China, we can conclude that the data is in line with the hybrid hypothesis.

## Discussion

Through systematic analysis of the oscillatory characteristics of 23 class B notifiable infectious diseases in mainland China from 2004 to 2020, three oscillation clusters (Figs 1 and 2), with different outbreak months, selectivity to specific month (Fig 3), and the change of oscillation strength with time evolution (Fig 4) were identified. The properties of each infectious disease are listed in Table 1.

### Comparison with previous works

To our knowledge, this is the first work to investigate the oscillatory properties of such a large number of infectious diseases in mainland China. Most previous works have included a single or a few similar diseases in China [23–26,43–45] or countries around the world [17–21]. Although some studies contained more infectious diseases [46], they did not systematically

investigate their oscillatory characteristics over time. We studied most of the infectious diseases (Class B) in mainland China from the perspective of the oscillation system and constructed a unified analytic framework to facilitate comparison. We also presented a method to categorize these diseases (Fig 2A). As illustrated with spectrum analysis, different diseases have different peak periods, different preferred outbreak months, and some have different selectivity. Diseases outbreak in autumn and winter are more selective, while those in summer and spring are less so. This finding will increase the understanding of the regularity of the diseases and guide in epidemic prevention.

Importantly, some infectious diseases, like HBV [47], HCV [48], HEV [49,50], Anthrax [51], Gonorrhea [52,53], Treponema pallidum [54], and Leptospirosis [55,56] are thought to be more sporadic rather than seasonal or cyclical. In our work, we found that they have distinct oscillatory properties despite relatively lower selectivity compared with other seasonal diseases.

## The trend of the epidemic situation in the mainland of China

In terms of the basic descriptive statistics of the 16-year time period investigated, the cases of infection of 14 out of 23 infectious diseases decreased over time, and nine out of 23 increased (Fig 4A and 4B). This shows that the control and preventative measures of the Chinese government have had a positive effect in these years. Moreover, we also found that the increase in oscillation strength often accompanies an increase in the number of infected cases, which will play an important role in the evaluation of epidemics in the future. Due to the typical cyclical fluctuation of the epidemic, the number of infected cases of one specific infectious disease in a month cannot reflect the real situation of the epidemic. If we take the hypothetical example that the average number of people infected with a certain disease over the past year is very large, this disease may even show a cyclical pattern every year, however, in one given month, only a few people are infected. Based on this, could we then assume that the epidemic has been effectively controlled and the peak has passed? The answer is no. As the results of our work, although this certain month is likely to be close to the peak of the epidemic cycle, the epidemic will rebound significantly in its preferred month, which then needs to be observed. People need to be more careful, and the government needs to strengthen its prevention and control during this period.

## Mechanisms of the periodic outbreak of infectious diseases in mainland China

The oscillatory properties of infectious diseases may be influenced by natural [27,29] or human factors [30–34]. In our results, Type I infectious diseases with relatively high selectivity in one year can be assumed to be seasonal. Natural factors, such as rainfall, temperature, and humidity, may affect the host. Some of the Type I infectious diseases with relatively low selectivity, are sexually transmitted, such as gonorrhea and treponema pallidum. The spread of these diseases may not be driven by natural factors, but by human behavior. However, in our results, these seemingly sporadic infectious diseases also have a clear periodicity, and the mechanism is still unknown. Type II infectious diseases (e.g., hemorrhagic and scarlet fever) have outbreaks in both summer (June) and winter (December) (Fig 2C). The intrinsic mechanism remains unclear and calls for further exploration. Type II infectious diseases (AIDS, HBV, and HCV) have relatively low selectivity to certain months or seasons. The cause of their annual outbreak frequency (three times per year) needs further investigation.

## Limitations of the current study

The primary limitation of this work is that we can only illustrate the properties under investigation descriptively. However, the oscillatory properties of infectious diseases reflect the

dynamic relationship among humans, pathogens, and the global environment. A future study should investigate these characteristics in a more detailed manner, and subdivide the underlying oscillation properties of infectious diseases using mathematical models [22,41,57,58]. Our results may inspire future modeling work to further explore the mechanisms of the recurrent outbreaks of infectious diseases.

## Supporting information

**S1 Fig.**
(EPS)

**S2 Fig.**
(EPS)

**S3 Fig.**
(EPS)

**S4 Fig.**
(EPS)

**S5 Fig.**
(EPS)

**S6 Fig.**
(EPS)

## Acknowledgments

We acknowledge National Health Commission of China for public dataset on cases of 23 class B infectious diseases.

## Author Contributions

**Conceptualization:** Chuanliang Han, Meijia Li, Xixi Zhao.

**Data curation:** Chuanliang Han.

**Formal analysis:** Chuanliang Han.

**Funding acquisition:** Xixi Zhao.

**Investigation:** Chuanliang Han, Meijia Li, Xixi Zhao.

**Methodology:** Chuanliang Han.

**Resources:** Chuanliang Han.

**Validation:** Chuanliang Han, Meijia Li, Xixi Zhao.

**Visualization:** Chuanliang Han, Meijia Li, Xixi Zhao.

**Writing – original draft:** Chuanliang Han, Meijia Li.

**Writing – review & editing:** Chuanliang Han, Meijia Li, Naem Haihambo, Yu Cao, Xixi Zhao.

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
