## [Decision Letter · Decision Letter 0]

3 Mar 2021

PONE-D-21-03876

Enlightenment on oscillatory properties of 23 class B notifiable infectious diseases in the mainland of China from 2004 to 2020

PLOS ONE

Dear Dr. Han,

Thank you for submitting your manuscript to PLOS ONE. After careful consideration, we feel that it has merit but does not fully meet PLOS ONE’s publication criteria as it currently stands. Therefore, we invite you to submit a revised version of the manuscript that addresses the points raised during the review process. Of greatest concern are reviewer #1's comments that the article is not presented in an intelligible fashion and is not written in standard English and reviewer #2's observations that the statistics and other analyses are not described in sufficient detail.

We look forward to receiving your revised manuscript.

Kind regards,

Nicholas S. Duesbery, PhD

Academic Editor

PLOS ONE

Journal Requirements:

Reviewers' comments:

Reviewer's Responses to Questions

**Comments to the Author**

1. Is the manuscript technically sound, and do the data support the conclusions?

Reviewer #1: Partly

Reviewer #2: Partly

2. Has the statistical analysis been performed appropriately and rigorously? 

Reviewer #1: I Don't Know

Reviewer #2: Yes

3. Have the authors made all data underlying the findings in their manuscript fully available?

Reviewer #1: Yes

Reviewer #2: Yes

4. Is the manuscript presented in an intelligible fashion and written in standard English?

Reviewer #1: No

Reviewer #2: Yes

5. Review Comments to the Author

Reviewer #1: I set out to review this submitted draft from a statistical perspective. However, the manuscript will first require substantial review and improvement to the grammar and wording. In its current form, the main text as well as figures/tables are somewhat difficult to follow. A few example include:

(1) Line 172 -- both components labeled "multiplication mechanism"

(2) Figure 3(C) -- is the Y-axis months of the year? What do the bars represent?

(3) Results section restates, word-for-word, descriptions from the methods section (e.g., paragraph starting at line 232)

(4) Table 1 -- make clear what is in last 2 columns (change in mean from first 8 years to second 8 years of study?)

These are only a small sample of the issues regarding intelligible writing. There are grammar and wording issues throughout the body of the manuscript not highlighted in this review. The authors should seek a colleague or external source with expertise in technical/scientific writing to review the manuscript prior to resubmission.

Additionally, the length of the manuscript could be reduced by concisely discussing the main purposes and findings of the analysis. Focus should be on the general findings across the 23 identified diseases. Describing characteristics of specific diseases becomes confusing for the reader, as the authors jump from disease to disease in the figures and results section. I recommend including disease-specific findings in Table 1 and in the supplementary figures only. The results section and main figures should present and describe the results of the analysis of trends across the included diseases.

Reviewer #2: This was a very interesting paper to review, and was quite thought provoking.

Introduction:

Methods:

To the understanding of this reviewer, spectral analysis of epidemiological phenomena is relatively uncommon. Seasonal decomposition, ARIMA modelling, and other methods are far more prevalent in the cited literature. A quick sampling of the bibliography yielded only a few studies employing this method and those appeared to focus on Maximal Entropy. For reproducibility, and ease of understanding for readers, I would recommend the authors describe the methods in more detail. For example, the calculation for power ratio is not given. Seeing this formula originates in electrophysiological research, it may difficult for an epidemiological reader to understand.

Lines 170 - 173: There appears to be a typo, both addition and multiplication result in the multiplication mechanism?

Results:

Line 238: Why was 0.5 selected as the cutoff?

Line 246: A non-parametric correlation coefficient would be more appropriate given the data distribution

Line 263-264: For readers not familiar with the interventions of the Chinese Government, this conclusion appears speculative. Please remove or reinforce with specific interventions.

Figure 3C is difficult to glean specific months from. Also, I feel there is a missed opportunity to colour the bars the corresponding disease classification (Type I, II, or III). This can provide more context.

Lines 308-316: This is difficult to follow. Are you concluding that simulated data with Gaussian noise following a hybrid model is similar to the observed data, thus proving the hybrid model? Why not calculate residuals or goodness of fit? Is that not a more plausible approach to validating the hybrid hypothesis?

Conclusion/Discussion:

Line 387: please reword

Overall Comments:

From the perspective of this reviewer, this appears to be a unique approach to assessing epidemiological time series. It was very interesting to read. However, at times the methods seem so similar to standard time series analysis (Seasonal Decomposition, Exponential Smoothing, etc) that I feel an explanation is needed as to why these more standard methods were not applied, or at least contrasted against.

6. PLOS authors have the option to publish the peer review history of their article (what does this mean?). If published, this will include your full peer review and any attached files.

Reviewer #1: No

Reviewer #2: No

---

## [Author Response · Author response to Decision Letter 0]

7 Apr 2021

Dear editor,

We would like to extend our utmost gratitude for your consideration, for securing 2 reviews for our manuscript and giving us this opportunity to revise our paper. We appreciate the comments and helpful suggestions from all reviewers. Based on these thoughtful suggestions, we have made substantial revisions for all parts of the paper. Addressing the comments has improved and clarified the communication of the findings we reported in our manuscript. We hope that our responses and revision are satisfactory, and hope this version will be more apt in addressing the requirements of the journal. Thank you so much for your time and consideration.

Below, we have humbly given point-by-point responses (in bold letters) to your comments (in italics and underlined). The changes in the manuscript were tracked for easy identification. 

Sincerely, 

Chuanliang

 

Editor 

https://journals.plos.org/plosone/s/file?id=wjVg/PLOSOne_formatting_sample_main_body.pdf
https://journals.plos.org/plosone/s/file?id=ba62/PLOSOne_formatting_sample_title_authors_affiliations.pdf

We have checked that our manuscript meets PLOS ONE's style requirements.

The minimal dataset underlying the results described in our manuscript can be found https://github.com/Stellapros/Dataset-of-infectious-disease

 

Reviewer #1: I set out to review this submitted draft from a statistical perspective. However, the manuscript will first require substantial review and improvement to the grammar and wording. In its current form, the main text as well as figures/tables are somewhat difficult to follow. 

We apologized for the grammar and wording issues, and a native speaker has helped proofread the paper and revised this manuscript.

A few example include: 

(1) Line 172 -- both components labeled "multiplication mechanism"

Thanks for your careful reading. We have modified it. (Please see line 175-180)

(2) Figure 3(C) -- is the Y-axis months of the year? What do the bars represent?

The Y-axis is the preferred month for each infectious disease, which is represented by the bars (The higher the bar, the closer to December, the smaller the bar, the closer to January) (Please see Fig S5). We have also clarified this in the manuscript.

(3) Results section restates, word-for-word, descriptions from the methods section (e.g., paragraph starting at line 232)

Thankk you for your careful reading, we have modified this section. (Please see line 236-241)

(4) Table 1 -- make clear what is in last 2 columns (change in mean from first 8 years to second 8 years of study?)

We have modified the title of the last 2 columns. Now it is ´Change in mean from first 8 years to second 8 years of study´ (Please see table 1). 

These are only a small sample of the issues regarding intelligible writing. There are grammar and wording issues throughout the body of the manuscript not highlighted in this review. The authors should seek a colleague or external source with expertise in technical/scientific writing to review the manuscript prior to resubmission.

Thank you for this kind suggestion. Upon further reading, we acknowledge this problem. A native speaker has proofread the paper and revised this manuscript.

Additionally, the length of the manuscript could be reduced by concisely discussing the main purposes and findings of the analysis. Focus should be on the general findings across the 23 identified diseases. Describing characteristics of specific diseases becomes confusing for the reader, as the authors jump from disease to disease in the figures and results section. I recommend including disease-specific findings in Table 1 and in the supplementary figures only. The results section and main figures should present and describe the results of the analysis of trends across the included diseases.

We really appreciate your suggestion and it makes the manuscript much clearer than before. We have reorganized the results and figures to focus on the general findings across the 23 identified diseases. The description of characteristics of specific diseases has been put into the supplementary section to avoid confusing of readers. (Please see results section)

Reviewer #2: This was a very interesting paper to review, and was quite thought provoking.

Thank you for the positive comment! It is very encouraging and we are happy to consider and address any issues you have.

Introduction: Methods:

To the understanding of this reviewer, spectral analysis of epidemiological phenomena is relatively uncommon. Seasonal decomposition, ARIMA modelling, and other methods are far more prevalent in the cited literature. A quick sampling of the bibliography yielded only a few studies employing this method and those appeared to focus on Maximal Entropy. For reproducibility, and ease of understanding for readers, I would recommend the authors describe the methods in more detail. For example, the calculation for power ratio is not given. Seeing this formula originates in electrophysiological research, it may difficult for an epidemiological reader to understand.

Thank you for your comments! 

We agree that perhaps more detail was needed. Additionally, we would like to add that the method of spectrum analysis has been used in a number of classic and modern studies in the field of infectious diseases to capture oscillatory strength[1–5]. This analysis is not limited to the electrophysiological research, but has a wider application to a variety of biological signals. We have described the methods in more detail (e.g., the calculation for power ratio) (Please see line 123-129，145-147)

Lines 170 - 173: There appears to be a typo, both addition and multiplication result in the multiplication mechanism?

Thanks for your careful reading, we have modified it. (Please see line 175-180)

Results:

Line 238: Why was 0.5 selected as the cutoff?

Thank you for this thought-provoking observation. After reflective consideration, we realized that the setting of this 0.5 is too subjective, so, we decided not to set a threshold to classify selectivity. Nonetheless, this cutoff does not affect the main results of this paper.

Line 246: A non-parametric correlation coefficient would be more appropriate given the data distribution

Thank you for this great suggestion! We have conducted a non-parametric correlation analysis and the significance is very similar compared with the parametric correlation coefficient (Please see line 243).

Line 263-264: For readers not familiar with the interventions of the Chinese Government, this conclusion appears speculative. Please remove or reinforce with specific interventions.

Thanks for your suggestion! We have removed these speculative words.

Figure 3C is difficult to glean specific months from. Also, I feel there is a missed opportunity to colour the bars the corresponding disease classification (Type I, II, or III). This can provide more context.

Thanks for your suggestion! Based on the suggestion of reviewer #1, we moved the original fig. 3 to the supplementary figures (Please see fig. S5). We have replaced the numbers with the name of each month for better presentation. We also colored the bars by the corresponding disease classification for better visualisation. 

Lines 308-316: This is difficult to follow. Are you concluding that simulated data with Gaussian noise following a hybrid model is similar to the observed data, thus proving the hybrid model? Why not calculate residuals or goodness of fit? Is that not a more plausible approach to validating the hybrid hypothesis?

We apologize for the unclear statement on the simulation part and thank you for your great suggestions. The simulated data based on Gaussian noise following the hybrid model is more similar to the observed data compared with the two other models. We realised that this is not enough to come to a conclusion, so, we fit the observed data using the three models (addition, multiplication, and hybrid) (Fig. 5). We found that the goodness of fit for the hybrid model is significantly larger than that of the other two models, which supports the hybrid hypothesis. (Please see line 186-201, 302-307)

Conclusion/Discussion:

Line 387: please reword

We appreciate this suggestion and have reworded. (Please see line 375-378)

Overall Comments:

From the perspective of this reviewer, this appears to be a unique approach to assessing epidemiological time series. It was very interesting to read. 

Thank you for the positive comment!

However, at times the methods seem so similar to standard time series analysis (Seasonal Decomposition, Exponential Smoothing, etc) that I feel an explanation is needed as to why these more standard methods were not applied, or at least contrasted against.

The aim of this work is to investigate the oscillatory property of infectious diseases. We considered that spectrum analysis is a more appropriate and more direct way of investigating these properties compared to methods on the time series analysis. The calculation of the power spectrum in each disease is based on the Fourier transform, which assumes that any time series can be represented as the summation of many periodical functions like trigonometric function (sin or cos). Using this method, we could directly investigate the frequency of the oscillations and the possibility of multiple oscillatory peaks in a disease. This differs from the idea of the standard time series analysis like seasonal decomposition. To give readers a clearer understanding of this analysis, we have added more descriptions in the manuscript. 

Reference:

1. Anderson RM, Grenfell BT, May RM. Oscillatory fluctuations in the incidence of infectious disease and the impact of vaccination: Time series analysis. J Hyg (Lond). 1984;93: 587–608. doi:10.1017/S0022172400065177

2. Broutin H, Guégan JF, Elguero E, Simondon F, Cazelles B. Large-scale comparative analysis of pertussis population dynamics: Periodicity, synchrony, and impact of vaccination. Am J Epidemiol. 2005;161: 1159–1167. doi:10.1093/aje/kwi141

3. Sumi A, Kamo KI, Ohtomo N, Kobayashi N. Study of the effect of vaccination on periodic structures of measles epidemics in Japan. Microbiol Immunol. 2007;51: 805–814. doi:10.1111/j.1348-0421.2007.tb03976.x

4. Greer M, Saha R, Gogliettino A, Yu C, Zollo-Venecek K. Emergence of oscillations in a simple epidemic model with demographic data. R Soc Open Sci. 2020;7. doi:10.1098/rsos.191187

5. Pons-Salort M, Grassly NC. Serotype-specific immunity explains the incidence of diseases caused by human enteroviruses. Science (80- ). 2018;361: 800–803. doi:10.1126/science.aat6777

---

## [Decision Letter · Decision Letter 1]

12 May 2021

PONE-D-21-03876R1

Enlightenment on oscillatory properties of 23 class B notifiable infectious diseases in the mainland of China from 2004 to 2020

PLOS ONE

Dear Dr. Han,

Thank you for submitting your manuscript to PLOS ONE. After careful consideration, we feel that it has merit but does not fully meet PLOS ONE’s publication criteria as it currently stands. Therefore, we invite you to submit a revised version of the manuscript that addresses the points raised during the review process.

Specifically, we ask that you address the minor revisions suggested by reviewer #2.

We look forward to receiving your revised manuscript.

Kind regards,

Nicholas S. Duesbery, PhD

Academic Editor

PLOS ONE

Journal Requirements:

Reviewers' comments:

Reviewer's Responses to Questions

**Comments to the Author**

1. If the authors have adequately addressed your comments raised in a previous round of review and you feel that this manuscript is now acceptable for publication, you may indicate that here to bypass the “Comments to the Author” section, enter your conflict of interest statement in the “Confidential to Editor” section, and submit your "Accept" recommendation.

Reviewer #1: All comments have been addressed

Reviewer #2: All comments have been addressed

2. Is the manuscript technically sound, and do the data support the conclusions?

Reviewer #1: Yes

Reviewer #2: Yes

3. Has the statistical analysis been performed appropriately and rigorously? 

Reviewer #1: Yes

Reviewer #2: Yes

4. Have the authors made all data underlying the findings in their manuscript fully available?

Reviewer #1: Yes

Reviewer #2: Yes

5. Is the manuscript presented in an intelligible fashion and written in standard English?

Reviewer #1: Yes

Reviewer #2: Yes

6. Review Comments to the Author

Reviewer #1: Thank you for responding to all reviewer comments and editing the manuscript accordingly. I find the manuscript now fit for publication.

Reviewer #2: This reviewer acknowledges and appreciates the revisions made to manuscript by the authors. Upon reading the section on goodness of fit, a small typo was noticed (Line 198-200). The line should perhaps be read as: "The goodness of fit for the above models is defined in Eq. 4." In addition, the authors go into detail about each component in equations 1,2, and 3; it would be useful for the reader to have the same level of exposition outlining the goodness of fit equation provided.

The remaining revisions are satisfactory, thank you for your contributions.

7. PLOS authors have the option to publish the peer review history of their article (what does this mean?). If published, this will include your full peer review and any attached files.

Reviewer #1: No

Reviewer #2: No

---

## [Author Response · Author response to Decision Letter 1]

13 May 2021

Dear editor,

We deeply appreciate the positive feedback from the reviewers! Addressed the comment from #2. Thank you so much for your time and consideration in making our manuscript the best it can be and we are delighted at the acceptance for publication.

Below are our point-by-point responses (in bold letters) to the reviewers’ comments (in italics and underlined). 

Sincerely,

Chuanliang

 

Reviewer #1: Thank you for responding to all reviewer comments and editing the manuscript accordingly. I find the manuscript now fit for publication.

Thanks for your support!

Reviewer #2: This reviewer acknowledges and appreciates the revisions made to manuscript by the authors. Upon reading the section on goodness of fit, a small typo was noticed (Line 198-200). The line should perhaps be read as: "The goodness of fit for the above models is defined in Eq. 4." In addition, the authors go into detail about each component in equations 1,2, and 3; it would be useful for the reader to have the same level of exposition outlining the goodness of fit equation provided.

Thank you for your careful reading and helpful suggestions! We have modified the typo (Please see line 197 in the Manuscript file) and added descriptions of the parameters and variables in Eq. 4 for a better reading (Please see line 201-203 in the Manuscript file).

---

## [Editor Report · Decision Letter 2]

24 May 2021

Enlightenment on oscillatory properties of 23 class B notifiable infectious diseases in the mainland of China from 2004 to 2020

PONE-D-21-03876R2

Dear Dr. Han,

We’re pleased to inform you that your manuscript has been judged scientifically suitable for publication and will be formally accepted for publication once it meets all outstanding technical requirements.

Kind regards,

Nicholas S. Duesbery, PhD

Academic Editor

PLOS ONE
---

## [Editor Report · Acceptance letter]

28 May 2021

PONE-D-21-03876R2 

Enlightenment on oscillatory properties of 23 class B notifiable infectious diseases in the mainland of China from 2004 to 2020 

Dear Dr. Han:

I'm pleased to inform you that your manuscript has been deemed suitable for publication in PLOS ONE. Congratulations! Your manuscript is now with our production department. 

Kind regards, 

on behalf of

Dr. Nicholas S. Duesbery 

Academic Editor

PLOS ONE